# Emotion Regulation Flexibility and Electronic Patient-Reported Outcomes: A Framework for Understanding Symptoms and Affect Dynamics in Pediatric Psycho-Oncology

**DOI:** 10.3390/cancers14163874

**Published:** 2022-08-11

**Authors:** Kasra Mirzaie, Anna Burns-Gebhart, Marcel Meyerheim, Annette Sander, Norbert Graf

**Affiliations:** 1Clinic for Pediatric Hematology and Oncology, Hannover Medical School, 30625 Hannover, Germany; 2University Outpatient Clinic for Children and Adolescents, Department of Psychology, University of Hildesheim, 31141 Hildesheim, Germany; 3Clinic of Pediatric Oncology and Hematology, Saarland University Hospital and Faculty of Medicine, Saarland University, 66421 Homburg, Germany

**Keywords:** emotion regulation, affect dynamics, electronic patient-reported outcomes, early warning signals, dynamical systems theory, pediatric psycho-oncology, pediatric cancer, adolescents and young adults

## Abstract

**Simple Summary:**

The individual’s ability to conceive and regulate the broad spectrum of their human emotions is closely linked to their mental health. The implications of a serious disease such as cancer represent an extraordinary burden to these internal coping mechanisms, especially in the case of young patients. Regarding their well-being and support, it is therefore of particular interest for caregivers to be able to follow the dynamics of the patient’s emotional world and perceptions. Technical progress enables new possibilities for data collection through tools for digital patient self-reports while simultaneously creating new challenges. Within the scope of this article, we provide an overview of the literature on this topic, outlining the current strengths and weaknesses and possible perspectives on digital aids, especially in terms of capturing the flexibility, fluctuations and early detection of symptom changes.

**Abstract:**

Emotion dysregulation is regarded as a driving mechanism for the development of mental health problems and psychopathology. The role of emotion regulation (ER) in the management of cancer distress and quality of life (QoL) has recently been recognized in psycho-oncology. The latest technological advances afford ways to assess ER, affective experiences and QoL in child, adolescent and young adult (CAYA) cancer patients through electronic patient-reported outcomes (ePRO) in their daily environment in real-time. Such tools facilitate ways to study the dynamics of affect and the flexibility of ER. However, technological advancement is not risk-free. We critically review the literature on ePRO in cancer existing models of ER in pediatric psycho-oncology and analyze strength, weaknesses, opportunities and threats of ePRO with a focus on CAYA cancer research and care. Supported by personal study-based experiences, this narrative review serves as a foundation to propose a novel methodological and metatheoretical framework based on: (a) an extended notion of ER, which includes its dynamic, adaptive and flexible nature and focuses on processes and conditions rather than fixed categorical strategies; (b) ePRO as a means to measure emotion regulation flexibility and affect dynamics; (c) identifying early warning signals for symptom change via ePRO and building forecasting models using dynamical systems theory.

## 1. Introduction

Cancer diagnosis and survival are often accompanied by high levels of distress, overall psychosocial impairments as well as various mental disorders, such as depression, anxiety disorders or post-traumatic stress disorders [1,2]. This holds particularly true for vulnerable groups, such as children, adolescents and young adults (CAYA) [3,4,5]. Furthermore, research in developmental psychopathology increasingly shows that emotion regulation (ER) in particular appears to be a relevant transdiagnostic factor and hence a necessary condition and potential mechanism for the emergence and maintenance of psychopathology across the entire age range including CAYA [6,7,8,9,10,11,12]. A similar trend is observable in pediatric psycho-oncology. Emotion regulation also appears to play a fundamental role in psychosocial adjustment for patients with cancer [13]. These insights have led to newly and specifically developed frameworks of emotion regulation in affective sciences and psycho-oncology across all age trajectories [14,15,16]. These are particularly, but not exclusively, relevant for CAYA because these age groups are confronted with different developmental tasks, possess varying baseline cognitive and emotional competencies and social support networks and thus varying resources and problem solving abilities compared to adults [17,18,19]. Distinct developmental and age-specific adaptations are, for instance, neurobiological maturation processes (such as synaptic pruning) that alter information processing and general cognitive abilities. Adolescence is also characterized by an overall higher emotional reactivity, especially toward negative emotions [20]. This often results in increased mood changes and distinct patterns of affectivity [21]. Furthermore, self-esteem drops to a nadir from the transition of childhood to adolescence, which is followed again by an increase during the transition to young adulthood [22]. In addition, adolescents experience what Elkind (1967) terms adolescent egocentrism or personal fable, in which an increased focus on one’s own appearance and beliefs about one’s uniqueness prevail while the social landscape changes [23]. These structural–functional constraints result in normative developmental tasks that must be overcome. Developmental tasks, such as vocational and financial independence, acceptance of one’s own body and its effective use or detachment and emotional independence from parents, are just a few examples of developmental tasks that are uniquely relevant to the psychosocial well-being of CAYA cancer patients. The occurrence of a life-threatening event and condition such as cancer in a population at the very beginning of their lives, results in idiosyncratic developments of pathogenesis, needs, treatment, coping and salutogenesis [24].

Simultaneously, due to various reasons, such as a rise in internet and smartphone accessibility worldwide, but also due to global circumstances such as the COVID-19 pandemic, digital health solutions (DHS) have become a valuable and much-needed resource for improving the quality of research and treatment of a wide variety of behaviors and diseases, especially within oncology and psycho-oncology [25,26,27]. They constitute a resource that inevitably must be reckoned with. These circumstances necessitate the exploration of their potential merits and detriments and thereby enable the innovation and advancement of psycho-oncology- and quality of life (QoL)-related research and treatment in the field of pediatric cancer. As we shall see, at the core of these DHS are electronic patient-reported outcomes (ePRO).

The aim of this article is to shed light on the mutual and interdependent relationship between emotion regulation and psychological distress in pediatric psycho-oncology and the role of ePRO in modelling those relations. In particular, challenges and benefits of ePRO in pediatric psycho-oncology as well as the resulting possibilities of ePRO use for the assessment of strategies, conditions, processes and the dynamics of ER and affective experiences in CAYA cancer patients will be explored while incorporating clinical experiences from the MyPal4Kids study [28]. Furthermore, empirical findings will be critically reviewed (a) to reciprocally inform and integrate both theoretical models and methodological approaches and (b) to formulate an argument for a potential overarching theoretical and methodological framework for using ePRO in psycho-oncology, especially in pediatric cancer patients. Overall, this paper aims to address relevant theoretical and methodological gaps and propose starting points and solutions for further investigations to researchers and scientist practitioners within the field of psycho-oncology.

## 2. What Are Electronic Patient-Reported Outcomes?

Electronic patient-reported outcomes are fine-grained data actively generated by patients on their current state [29]. In psychological research, ePRO are often referred to as experience-sampling methods or ecological momentary assessment. These methods can be subsumed under the notion of intensive longitudinal data and understood as (actively or passively) repeated measures and real-time samplings of patients’ current physical or psychological states mostly within their natural environments [30,31]. Active data sampling occurs primarily through patients’ self-reports via apps on electronic devices. Questionnaires and diary functions can be built into such apps to be answered by patients in real time. These samplings can be signal- or event-based. Signal-based sampling means that patients are reminded and prompted by the app to report on their conditions at certain fixed or random times during the day. Event-based sampling means that patients can provide information about their conditions without solicitation and of their own accord. These reports are often made before, during or after occurring events, such as daily hassles. In the case of CAYA, caregivers or third parties, such as health care professionals (HCPs), may in addition externally assess current states. Passive data sampling occurs without the patients’ conscious involvement for certain observational or physiological states (e.g., step counts, heart rate, etc.).

## 3. Electronic Patient-Reported Outcomes in Pediatric Cancer and Psycho-Oncology

The assumptions of ePRO benefits for pediatric psycho-oncology are not only based on the success and enhancement of real-time monitoring of treatment process and outcome in neighboring fields such as public health, psychiatry, clinical psychology and psychotherapy [32,33,34]; digital health and the clinical use of ePRO have also reached contemporary adult and pediatric oncology, palliative care and QoL-related research [35,36]. The importance of the assessment of QoL in cancer trajectory is shown in various systematic reviews and meta-analyses of randomized controlled trials. These reviews show that the assessment of QoL through QoL scales has beneficial prognostic validity and predictive power for cancer survival [37,38], although further research on explanatory mechanisms through which ePRO relate to survival is necessary [38]. Growing evidence suggests that the use of outcome measures in routine oncological care may improve well-being and enable improved patient-centered cancer care and overall personalized treatment [39]. In fact, the European Association for Palliative Care emphasizes the relevance and necessity of ePRO use in children and adult palliative care, especially for cross-country comparisons [40]. Furthermore, LeBlanc and Abernethy (2017) state that ePRO can help to detect unrecognized symptoms and unmet needs of this vulnerable patient group while simultaneously involving them more in the entire treatment process [41]. This seems paramount for CAYA patients since the entire family system and other support systems (e.g., peer groups) are involved in the treatment and care process [42,43]. Outcome measures seem imperative in CAYA palliative care since the nature and severity of these young patients’ conditions pose great demands for medical care and communication [42,44]. A recent review on evidence-based standards of care in pediatric psycho-oncology suggests that ePRO measures are promising for routine assessment in research and practice [24]. Another systematic review concluded that on average, HCP-patient communication is improved through DHS but that these new tools require further improvement and research [45]. A systematic review of Lau and colleagues (2020) found that eHealth interventions in youth with chronic illnesses show positive treatment responses. Regardless, more research is needed [46]. Similar evidence for eHealth interventions in CAYA cancer patients was found in another psycho-oncology review [47].

Especially in psycho-oncology, ePRO pose an advantageous method for monitoring the treatment process and supporting HCPs [48]. Furthermore, a study by Abrol et al. (2017) showed that a desire for digital resources exists in digital natives, such as teenagers and young adults with cancer and particularly in cancer survivors who report the need to connect to fellow patients [49]. The latter also applies to adult cancer patients [50]. The results of a multicenter study further suggest that ePRO may have great potential for in vivo data collection related to CAYA health and psychosocial development [51]. Bagot et al. (2018) further illustrate the importance and necessity of interdisciplinary research [51]. One way to collect data in children efficiently is through gamification via mobile apps, in which children can be immersed into a virtual gaming environment and occasionally report back current physical or psychological states [52]. Our European multicentered and transdisciplinary research project called MyPal has developed such a gamified DHS, a serious game [53]. In the MyPal4Kids study, we test the AquaScouts app at three pediatric oncology clinical sites in two European countries for usability, acceptability and feasibility of ePRO for palliative care in pediatric oncology patients and their HCPs [28]. Gamified DHS are expedient since they may counter reporting fatigue [53]. This is particularly relevant in the care and research of children and adolescents. Figure 1 shows a screenshot of the game AquaScouts along with associated items from the caregiver app.

One of the major factors for the successful treatment monitoring of psychological interventions as well as the acceptance, feasibility and willingness of patients to use DHS, such as ePRO, is obtaining feedback from HCPs [54,55,56]. Patients’ involvement in the reporting process and, with that, the frequency and quality of feedback conversations with the HCPs, highly influence the patients’ sense of utility and meaning [32]. Visual aids of developmental trajectories of symptomatology and of the assessment of context factors (such as medication, daily hassles, etc.) can help with feedback discussions [33]. Figure 2 depicts such trajectories of various cancer- and QoL-related questionnaire items as well as contextual information on the MyPal platform for HCPs. Preliminary results from the MyPal4Kids study show that meeting such patients’ needs for HCP feedback is fundamental for the acceptance, sense of valuation, benefit and ultimately for the compliance of high-frequency reporting (Meyerheim et al., in prep.).

From a psycho-oncological perspective, we propose the study of coping and emotion regulation as a crucial mechanism for psychosocial adjustment during cancer distress in CAYA. We will, in addition, make a converging argument for ePRO use in assessing and measuring affective experiences, ER and coping strategies and processes in pediatric psycho-oncology.

## 4. Current Emotion Regulation Frameworks in Psycho-Oncology

Coping with distress—as caused by death anxiety, for instance—forms one of the core principles of psycho-oncological treatment and palliative care [57]. Cancer patients are often not only in need of care but also of a sense of agency. Evidence shows that intense existential and emotional conditions in palliative care, such as death anxiety, can have significant impact on advanced end-of-life care-planning due to reduced communication [58]. In CAYA and adult cancer care, strategies such as emotional disclosure or acceptance [59,60] help patients make sense of such debilitating conditions, give back agency and regulate overwhelming emotions, thus rendering cancer bearable. These strategies can be understood within the framework of third-wave therapies—such as acceptance and commitment therapy [61] or dialectical behavior therapy [8,62]—as well as humanistic frameworks—such as person-centered therapy [63], logotherapy or existential and meaning-centered psychotherapy [64,65], among others [66,67,68]. Overall, coping and emotion regulation are at the core of psychosocial well-being. For this reason, researchers are increasingly scrutinizing the construct of emotion regulation within various research and treatment contexts, such as psychopathology, psycho-oncology and cancer distress [14,15,16,69].

In psychological research, various attempts have been made to classify an array of concrete emotion regulation strategies (ERS) into categories at a higher level. The most recognized classes are adaptive and maladaptive ERS [9]. A meta-analytical review by Aldao et al. (2010) focusing on ERS across psychopathology in adults and CAYA showed that six specific strategies are most commonly researched and used in ER-study designs and are most predictive of psychopathology: problem solving, rumination, reappraisal, avoidance, acceptance and suppression. These six strategies show a robust association with psychopathological symptoms and psychological well-being. In particular, they found that ERS problem solving, reappraisal and acceptance were negatively associated with psychopathology, meaning that individuals reporting to use such strategies were significantly less vulnerable to mental disorders. In contrast, rumination, avoidance and suppression were positively associated with psychopathological symptoms, meaning that individuals reporting to use these strategies significantly showed higher levels of distress symptomatology [9]. These findings led to the assumptions that there are overall ERS that are more beneficial to mental health and thus functional and adaptive, whereas in contrast, there are ERS that are detrimental to mental health and hence adverse and maladaptive. For instance, a systematic review and meta-analysis by Baziliansky and Cohen (2021), sought to investigate the relationship between emotion regulation patterns (i.e., adaptive and maladaptive ERS) and psychological distress in cancer survivors. The authors found that specific strategies were in fact associated with higher levels of psychological distress symptoms, but high variability in the association between ER patterns and distress exists. They conclude that further studies with consistent methodologies are required to investigate and recognize ER patterns and psychological distress [16].

A review by Conley et al. (2016) highlights the role and importance of ER across the cancer trajectory. The authors review principles of ER and their assessment and propose an ER model with mediational pathways for effects on psychosocial outcomes in cancer patients and survivors. A dual-route approach suggests a mediation of ER on psychosocial outcomes via engagement or disengagement strategies (i.e., approach strategies such as problem-solving or avoidance strategies). The authors further tested their proposed ER model empirically in a longitudinal study on mental-health-related QoL in patients with recurrent breast cancer. Their results show that higher baseline levels of negative emotions were associated with detrimental QoL at 12 months independent of strategy use, indicating that ER is in fact associated with QoL. However, the authors also stated that their applied diagnostic instrument was insufficient for the measurement of ER. Overall, Conley et al. (2016) presented a testable ER model in cancer based on strategy use. Although the testability of their model is viewed as a strength, their results show regulatory processes independent of the use of fixed strategies against what their model suggests. The authors conclude that affective regulatory processes are relevant for the entire spectrum of cancer survivorship but that long-term follow-up studies are necessary to depict affect dynamics and reciprocal effects of ER for cancer survivors [14].

Kangas and Gross (2020) review existing self-regulatory models in psycho-oncology that are mostly based on Lazarus and Folkman’s (1984) [70] coping theories, such as Leventhal et al.’s common sense model and others. The authors conclude that most common self-regulatory models of cancer distress are rather cognitively oriented. They report that there is little research investigating the relationship between emotional and cognitive processes and that existing models (e.g., [14]) do not account for both affect-generative and regulatory processes across the entire cancer trajectory. Therefore, the authors propose the Affect Regulation in Cancer Framework (ARC) that claims to close the discrepancy between emotional and cognitive processes across different stages of the cancer trajectory. The ARC distinguishes itself from other models by being process-oriented and therefore causal. This is characterized by a feedback cycle of different regulatory processes, such as attention deployment, cognitive change, response modulation, (cancer-specific) situational selection and modification. The cycle can be applied to all varying stages and phases of cancer, i.e., from the prediagnostic (screening) phase to the treatment phase to the end-of-life (terminal) phase. Explaining such processes at different phases is the main strength of the ARC. Furthermore, the ARC tries to break away from classical and categorical ERS. However, the authors equate strategies with processes. This poses as a serious weakness of the framework since it is yet unclear how much autoregulatory processes (meaning unconscious and visceral self-organization at different subpersonal levels, such as physiological, neurobiological, computational, etc.) account for ER and how much of ER can be explained by active and conscious use of ERS. Furthermore, the authors disclose that their framework has not been empirically tested yet [15]. In line with Brandao et al. (2016) [13], the authors conclude that multiphase longitudinal research is required. Given that the model is new and only recently published, pending empirical testing is necessary. Furthermore, despite the lack of empirical data and conceptual completeness, the ARC appears to be a useful heuristic model with potential to account for recursive processes and causal events at different phases [15].

Considering the limitations of the aforementioned models, we argue in favor of a conceptual adjustment and propose a model that goes beyond the simple use of strategies and their fixed categorical representation; regulatory processes and situational contexts should be included.

## 5. Emotion Regulation Flexibility and Affect Dynamics: An Alternative Model

In comparison to the already established ER models, we would like to present an alternative model that necessarily and sufficiently captures the flexible use of ERS, the dynamic processes of ER and affect, as well as their underlying conditions. We further propose this alternative model for the use in psycho-oncology and ePRO research.

According to Barnow et al. (2020), ER models that divide ERS into adaptive and maladaptive are outdated since the appropriateness and inappropriateness of one’s ability to regulate emotions is highly context-dependent [71]. ERS are categorized into adaptive and maladaptive on the basis of their correlative relationship towards psychopathological symptoms and well-being [9]. The implications derived from such correlations are of statistical nature and pose justifiable risk factors for mental health [72,73]. For instance, tendency to ruminate or suppress emotions is positively associated with psychological distress symptoms in cancer survivors [16]. Rumination and experiential avoidance and their association with depressive symptoms and post-traumatic stress disorders can further be observed in parents of children with cancer [74,75]. However, such correlative studies do not allow for causal inferences since they do not provide any information about contextual relevance and boundary conditions. According to Barnow et al.’s (2020) model of emotion regulation flexibility, however, suppressing—for example—one’s nervousness during a job interview can be an adaptive and effective strategy. The same holds true for other ERS-like rumination [71]. There is evidence that positive cancer-related rumination is associated with post-traumatic growth [76]. Alternatively, they propose a model that accounts for the flexible and context-dependent use of different ERS as well as for their frequency and effectiveness. In detail, the authors identify five specific factors that are relevant to the deployment, development and maintenance of successful emotion regulation. According to them, the controllability of a situation, the social or interpersonal context, the emotional intensity, the number of stressors and the regulatory goals are necessary conditions in order to infer the adequacy or inadequacy of ERS use. Table 1 lists these five context-sensitive conditions and juxtaposes different regulatory strategies and their adaptiveness in typical and varying oncological settings from the patient’s perspective in order to highlight contextual relevance for the effectiveness of ERS in pediatric psycho-oncology. Similar ERS and ER processes apply to caregivers such as parents. This is particularly important for psycho-oncology, since different cancer phases and stages mean varying affordances and salience landscapes. What information discloses itself to us as relevant is at the core of our agency [77]. This translates into an altered access to and to varied sets of ER skills and strategies depending on the cancer phase. In contrast to the ARC, the model of emotion regulation flexibility may help explain such emerging dynamics of ER and affect across the developmental trajectory by accounting for context. The context-sensitive conditions may be applied to any cancer phase.

It is, however, important to note that the model of Barnow and colleagues (2020), to our knowledge, has not yet been empirically tested in an oncological setting. Although supporting empirical evidence exists for ER variability and flexibility and their relationship towards affective experiences (negative affect) in young adults, the model has yet to be further tested, especially in pediatric psycho-oncology and QoL-related research [78].

## 6. Assessing Emotion Regulation Strategies, Processes and Conditions through ePRO

Due to the nature and dynamics of uncertainty in oncology and psycho-oncology, ePRO can provide opportunities to investigate context-sensitive conditions necessary for various theoretical and causal assumptions. They are particularly relevant for the explanation and prediction of adjustment, adaptation and flexibility in emotion regulation [71,79,80].

As shown above, Barnow and colleagues (2020) emphasize their importance and propose a way to capture and measure these factors via ePRO [71]. Scales and items concerning these context factors can be digitally entered into the respective app (e.g., “How controllable did you find the situation?”). The assessment of these factors can then happen in real-time in real-life settings that evoke certain emotional states. Especially in pediatric oncology, where uncertainty is high and lots of interactional levels meet, such contextual factors and boundary conditions matter for understanding disease progression and planning interventions. The benefits of ePRO lie in the ecological validity of this type of approach. Furthermore, ePRO-based protocols, such as those from the MyPal-Adult [81] and MyPal4Kids [28] studies, can incorporate passive data sampling in form of integrated app functions, such as pedometers or heart rate monitoring via Wearables. This data collection technique and research methodology is also known as telemonitoring and often finds application in neighboring medical fields, such as cardiology or sports medicine. Passive data sampling may be exploited for studying autoregulative and visceral processes underlying ER- and QoL-related variables, such as physiological stress symptoms.

These approaches are conceptually congruent with the abovementioned models of emotion regulation within and outside the realm of psycho-oncology, especially with the ARC [15]. The ARC, according to Kangas and Gross (2020), is explicitly procedural (process model of emotion regulation) and therefore causal and explanatory [15]. Causal events alone, however, without their enabling and limiting constraints as well as contextual parameters, are insufficient for explanation [82]. This particularly holds true for the explanation of agency-related phenomena, such as ER [71,82]. Investigating constraints and conditions that make cause and effect possible are just as necessary for the explanation of ER and psychological well-being as their causal events. The investigation of the appropriateness of ERS therefore must be viewed under both factors, i.e., events and underlying conditions that evoke the use of certain ERS.

We will present a metatheoretical framework that accounts for complex interactions and dynamical processes as well as their underlying conditions. This framework bears further potential for forecasting models of psycho-oncological and QoL-related parameters using ePRO.

## 7. The Use of ePRO for Process Monitoring and Early Warning Signals: A Metatheory

ePRO can be exploited for symptom diagnostics, treatment process monitoring and forecasting models [83,84,85,86]. However, research methods and an overarching frame of reference are needed to identify and model early warning signals, tipping points and patterns of changes in various psychological and QoL-related variables, such as ER, mood and affect, distress, physiological and psychological symptoms, HCP–patient relationship and communication, therapy motivation, contentment and compliance [14,16,87]. Dynamical systems theory (DST) may provide such a language that gives meaning to ePRO and may thus function as such a metatheoretical framework [88]. Due to its transdisciplinary origin and understanding of emergence and self-organization, DST allows the investigation of and provides tools for modeling interacting complex systems and thus synchronous (vertical) and diachronic (horizontal) dynamic processes [17,89,90,91,92,93,94,95,96].

Schiepek et al. (2014) and others applied DST to the clinical field and developed a way to measure change processes [97,98]. They view clinical interventions such as psychotherapy or psycho-oncological treatment as a self-organizing process. In the language of DST, change processes and periods of destabilization in the intensity and quality of order parameters, such as symptoms, are referred to as phase transitions, whereas as stable states, processes and periods of consolidation can be referred to as attractors [98,99,100]. The dynamic and sudden shift from one state to the other is referred to as self-organized criticality [101]. In general, (symptom) stability can be understood as a special case of change [17]. The idea of attractors has further been applied to cancer networks and been reviewed through the lens of DST [102].

Recent empirical findings from studies on adults are indicative for the assumption of the role of change processes in various psychopathologies such as mood disorders or obsessive–compulsive disorders. These findings show that periods of unstable depressive symptoms in the context of psychotherapeutic treatment are associated with higher therapeutic success [98,103,104]. In several studies, Olthof et al. (2019; 2020) were able to show that critical fluctuations in the self-reporting of depressive symptoms and of the therapy process itself via ePRO preceded and predicted symptom severity, improvement and treatment success [103,104]. Schiepek et al. (2014) were able to show such results in patients with obsessive–compulsive disorders [98]. Similar findings on affect dynamics and their role in the development of prospective psychopathological symptoms have been shown in adolescents [105]. Another study also shows the connection between daily affect dynamics and emotion regulation in terms of emotion regulation flexibility in young adults [78]. The authors found that variable and flexible switching between different ERS was associated with a reduced negative affect experience. This is one of the first studies to elucidate the adequacy of ERS in relation to contextual demands, showing the daily variability and dynamics between ER and affect using ePRO.

Furthermore, the extent, intensity and quality of the affective states and thus the emotion regulation process itself may be relevant [106]. Empirical findings show that fluctuations in emotional and mood experiences (affect dynamics) themselves seem to have a decisive influence on the emergence, extent and course of psychopathological and distress symptoms [32,95,99,107,108,109]. In clinical research, time-intensive measurement methods (such as ePRO) have shown that a disequilibrium in affective experiences and symptoms precedes decisive changes to symptoms and distress levels [109,110,111]. Thus, personal affect dynamics may represent early warning signals for changes in symptomatology. There are different types of affect and symptom dynamics, each with idiosyncratic effects and structures. Looking at their trajectories, preceding processes with local/global maxima/minima called critical fluctuations or critical slowing-down serve as early warning signals for what are called “sudden gains and losses” in symptom severity [104,110,112,113,114,115,116]. Figure 3 shows a schematic representation of dynamic changes in symptom severity during treatment.

The strengths of these studies lie in the modeling of complexity in individuals’ time-intensive self-reports of well-being and monitoring of the treatment process [87]. The identification of early warning signals and predictors of symptom change can be accomplished by using time series or network analyses [117,118,119]. The use of network science in cancer research has already revealed fruitful and promising insights at the genomic and metastatic levels [102,120,121]. Similar findings exist using time series analyses for forecasting tumor growth [122,123]. These methods have further been successfully transferred to the behavioral and phenotypical level and been applied to the study of mental disorders and psychopathology in various age groups, including CAYA [105,124,125,126,127,128]. Schiepek et al. (2020), for instance, proposed various measures for the identification of early warning signals in time series [94,117]. Such measures can be used for idiographic analyses on the personal level and can be aggregated on the interpersonal level [93,129,130,131,132,133,134,135,136,137]. In particular, survival analyses are commonly used for forecasting models in cancer research [138,139,140,141,142]. The assumptions of DST have been widely accepted in research on psychopathology, and an understanding of mental disorders as complex (adaptive) systems is generally accepted [104,143,144,145,146,147].

Such evidence is not only found in research on psychopathology and psychotherapy but is also already evident in childhood cancer research. For instance, Meryk et al. (2022) presented a case study in favor of ePRO and early warning signals. In this study, the authors argued for a web-based approach for daily child self-report and its benefits for early symptom detection in an outpatient oncology setting. Using an adapted self-report measure from the Pediatric Quality of Life Inventory for the daily assessment of various symptom and QoL-related variables, they monitored a child with Burkitt leukemia over a period of 16 days, identifying different stages of symptom severity and facilitating real-time interventions. The authors argue for a promising use of ePRO for early symptom detection in daily clinical routine and for further clinical trials across whole patient populations [148].

A study conducted in an ambulatory oncology clinic by Watson and colleagues (2021) built such forecasting models using ePRO data for time series analyses (autoregressive integrated moving average; ARIMA) to forecast and manage the symptom complexity of adult cancer patients. The objective of the study was to examine the predictive power of ARIMA models on the percentage of cancer patients with high symptom severity. Using a symptom complexity algorithm, the authors allocated a variety of physiological and psychological symptoms by means of severity and number of concerns. The algorithm triaged symptoms into visual flags (green/low, yellow/moderate, red/high) and assigned a symptom complexity score. Over a period of 24 weeks, the authors collected ePRO data from completed self-report questionnaires and fitted them to their ARIMA model. After 24 weeks, they compared their predictive ARIMA model, which fitted historical data from the past 24 weeks, against observed data during another forecasting period of 8 weeks. Forecasting accuracy of the model was assessed using mean absolute predictive error (MAPE). Overall, they found an acceptable forecasting value of 5.9%, meaning that their model over- or under-reported on average 5.9% (1.9% to 11.8%) of high symptom complexity, which is acceptable and within the typical 5% MAPE threshold [86]. Similar findings are reported on time series forecasts using common-day clustering for outpatient clinic visits, ARIMA or Bayesian methods for incidences of different cancer types as well as forecasts for health-related QoL in breast cancer [149,150,151,152]. These types of research methodologies can further be used to evaluate intervention programs. Waller et al. (2010) used an interrupted time series design to evaluate the efficacy of needs-based palliative care intervention on young adult and adult cancer patients [153,154]. Similar methodologies using time series were applied in order to test psychosocial assessment, care and treatment and distress management for young adult and adult cancer patients [155].

## 8. Summary: Benefits and Drawbacks

ePRO represent technological advances in digital health solutions for CAYA oncology. Such advances come with a host of demands and ramifications. We present a summary of different merits and detriments of ePRO use in CAYA oncology. Bertucci et al. (2019) and Dinkel (2020) carried out an analysis of strengths, weaknesses, opportunities and threats (SWOT analysis) for the use of digital health solutions in oncology [35,156]. However, since ePRO constitute only a subset, unrelated points are omitted in favor of ePRO-based aspects specific to psychological interventions, psycho-oncological and QoL-related research and treatment in CAYA [32]. Table 2 summarizes strength, weaknesses, opportunities and threats of DHS and ePRO in CAYA psycho-oncology in a detailed fashion (adapted from [32,35,156]).

Overall, ePRO constitute a promising way to capture affective experiences, QoL and symptom severity among others at different stages on the cancer trajectory within and between patients. Furthermore, they can be used to identify change patterns in those mentioned factors. The ecological approach to fine-grained data collection and thus to potentially relevant information may assist in planning and adjusting necessary therapy protocols. Another major strength of ePRO is their potential to afford valuable communication channels and therefore to strengthen HCPs and patient communication.

However, ePRO can be burdensome for both patients and health care professionals if implemented under adverse conditions, especially if the infrastructures of health care providers do not allow for effective applications. Moreover, the use of and dependency on technology creates a host of serious ethical issues that need to be addressed in step with their technological advancements [28].

## 9. Conclusions

Emotion regulation (ER) plays a crucial role in psychosocial well-being as well as health-related QoL. Evidence shows, however, that ER is dynamic rather than static in nature. In order to further develop and incorporate ER-based interventions in psycho-oncology and palliative care, research designs are needed to investigate the structure and function of a flexible and dynamic ER used for pediatric and only after adjustment for other cancer patients as a result of prospective clinical trials. We propose different tools and frameworks for studying change, affect dynamics and emotion regulation flexibility. At the core of these tools are electronic patient-reported outcomes (ePRO). Study designs using ePRO can provide repeated measures and intensive longitudinal data, which can be further exploited for needs-based care protocols in psycho-oncology and QoL-related palliative care, e.g., in vulnerable patient groups like children, adolescents and young adults. This is currently being investigated in a European research project on palliative care called MyPal, which includes an adult as well as a child and adolescent study. Furthermore, ePRO data can be exploited for early warning signals and forecasting models for disease progression, symptom severity and management as well as QoL-related palliative care. However, a meta-theoretical framework, such as dynamical systems theory (DST) is necessary for the study of symptom dynamics and change processes. DST can provide tools for predictive analyses, such as time series and network analyses. However, further empirical research is necessary to investigate if such frameworks are sufficient. The proposed frameworks for dynamical and flexible ER are best tested using longitudinal approaches and randomized controlled trials in pediatric cancer settings. Furthermore, common ePRO methodologies as well as a shared set of evidence-based ePRO protocols, assessment instruments and technologies are required for comparability.

## Figures and Tables

**Figure 1 cancers-14-03874-f001:**
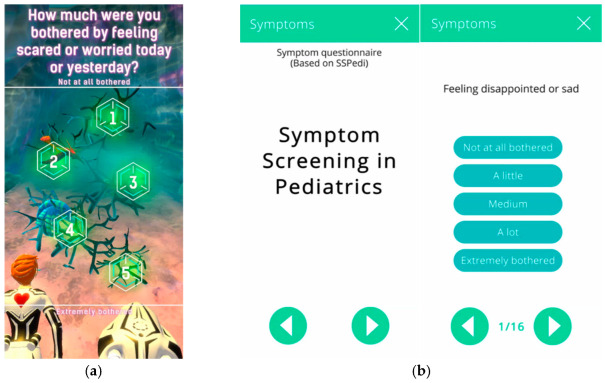
(**a**) Example of a question appearing in the MyPal Child App serious game; (**b**) example of a symptom question appearing in the MyPal Carer App, to be answered by proxy.

**Figure 2 cancers-14-03874-f002:**
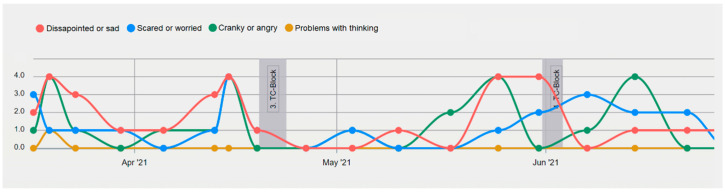
MyPal Symptom Trajectory. Example of symptoms reported by a patient undergoing chemotherapy treatment with topotecan/cyclophosphamide (TC-Block). This patient tended to show increased emotional distress symptoms prior to an upcoming chemotherapy block. Symptoms are graded according to their severity (not present to extremely strong).

**Figure 3 cancers-14-03874-f003:**
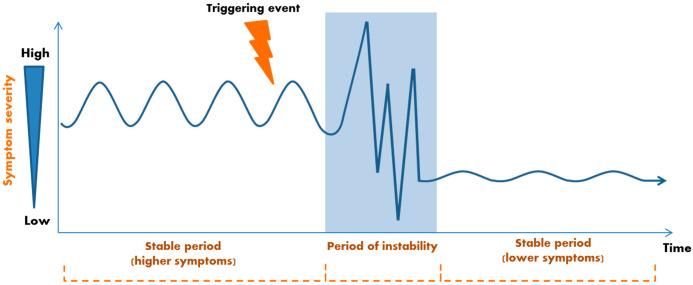
Schematic model: a variety of (internal or external) triggering events and stressors, such as treatment, can cause perturbations and periods of destabilization in the system and with that a qualitative shift or phase transition in symptom severity from a previously stable state. The system synchronizes and stabilizes itself after a period of instability at a qualitatively different and stable symptom level again (adapted from [104]).

**Table 1 cancers-14-03874-t001:** Examples of the potential adaptiveness of various emotion regulation strategies depending on five context factors. Situations A and B represent hypothetical examples of different oncological scenarios. Varying situations result in varying perceptions of these five factors (e.g., low vs. high degree of controllability), which in turn affect agency and afford the use of varying strategies. Different regulatory strategies are underlined. Patient’s introspections are in italics. In this table the underlined regulatory strategies are always adaptive (based on the model of emotion regulation flexibility [71]).

Context-Sensitive Condition	Adaptiveness of Emotion Regulation Strategies Depending on Contextual Relevance	
	Situation A: Low Degree	Situation B: High Degree
**Controllability within a situation**	Reframing and reappraisal during chemotherapy nausea (“*Nausea means therapy is working*.”)	Resilience and tenacity during convalescence and rehabilitation (“*I will keep pushing*!”)
**Interpersonal context**	Suppressing shame in front of physicians during rounds (“*Be courageous. The examination will be over any second*.”)	Emotional disclosure and social support when experiencing injustice (“*Why me? It makes me so angry! I want to talk to mom about this*.”)
**Emotional intensity**	Distraction and attentional shifting during puncture (“*I will look at the decorated wall and not at the syringe*.”)	Acceptance and tolerance of grief in palliative care (“*It is fine. Embracing sadness can be liberating*.”)
**Number of stressors**	Nonjudgmental awareness of emotions and mindfulness (“*My mind feels scattered. I will focus on my breathing for a moment*.”)	Problem-solving, such as scheduling demands during different stages (“*I will partake in the online class after physical therapy*.”)
**Regulatory goals**	Inhibiting negative emotions (“*I don’t want to feel nauseous after radiation today*.”)	Activating positive emotions while inhibiting negative emotions (“*I want to feel happy eating ice cream after radiation and not feel nauseous*.”)

**Table 2 cancers-14-03874-t002:** SWOT analysis of ePRO use in CAYA psycho-oncology. Abbreviations: HCP = health care professionals, ePRO = electronic patient-reported outcomes. Adapted from [32,35,156].

Strengths	Weaknesses	Opportunities	Threats
Involvement of patients into own therapy and monitoring process provides agency and emotional and self-regulation Strengthening HCP–patient communication through sustained feedback cycles Promoting the role of HCPs as “scientist practitioners” Accessibility of smart-devices and ecological validity due to data sampling in actual lived environment Reducing recall bias for self-report measures through real-time tracking ePRO useful for symptom management (e.g., visualizing symptom trajectories) Diary functions beneficial for providing contextual information on patient’s state Bypassing the burden of face-to-face interaction, particularly in young cancer patients where shame and timidity may play an adverse role in HCP–patient communication Affording participation by assuming and assigning identities other than “doctor–patient” Expression of appreciation for patients, their participation, perceptions and efforts Evaluation of treatment protocols Rapid access to patient records, second opinions, information about illness and treatment Facilitating transfer between therapy and everyday life Providing security, structure and routine through daily self-report rituals Sharing disease and treatment related experiences (forums, social network)	New and complex organizational forms and infrastructures that may clash with already existing structures Lack of coordination between practitioners Insufficient training in digital skills in patients and practitioners High demand on all sides of actively involved people (from parents and patients to HCPs) Digital health solutions such as serious games can cause discontent if not sufficiently attractive, which may hinder active participation Data security, ethical and bioethical issues Reducing communication to digital symptom reporting	Diary functions as a means for distress regulation through expressive writing and emotional disclosure Digital natives: ePRO particularly useful in adolescents who are overall under-studied due to lower compliance Potent tool for diagnostics, research and treatment in psycho-oncology where assessment for disease and affect dynamics is lacking Early detection and pattern identification of symptoms and behaviors at different scales using methods from dynamical systems theory Bypassing the problem of group-to-individual-generalizability using idiographic science and personalized care Real-time tracking enables enhanced state-trait research at different time scales Fine-grained data necessary for tools and algorithms for filtering signal from noise Enables the study of symptom severity in relation to cross-contextually variant/invariant factors (e.g., cancer-related fatigue at different day times) Increased cooperation between healthcare facilities Fostering multi-professional teamwork between different HCPs Emergence of new health care professions (e.g., nurse navigators) Opening up to the digital market	Potential iatrogenic effects of reminding and burdening the entire domestic environment with the issue of cancer Insufficient HCP feedback may lead to patients’ discontent and sense of futility, a negative return on investment and a feeling of being instrumentalized Lack of understanding associated risks in complex systems (e.g., delayed effects) Digital health solutions and ePRO may further contribute to digital divide Prompting can have paradox effects on agency, self-determination and the need for rest if requests for active participation are experienced as exhausting and as duty Increased screen-time may have effects on (a) parents’ openness to participate (b) patients’ motivational drive and (c) the desire for more interpersonal quality time on both ends, especially in palliative and end-of-life care Virtual, less “human” relationships Taking refuge in virtual worlds (virtual exodus) More data means more noise, which creates the need for relevance filters and dependency on algorithms Trivializing the burden of medical care

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
