# Peer review of "Emotion Regulation Flexibility and Electronic Patient-Reported Outcomes: A Framework for Understanding Symptoms and Affect Dynamics in Pediatric Psycho-Oncology"

_cancers, 2022, doi:10.3390/cancers14163874_

Round 1

Reviewer 1 Report

This review is a comprehensive work on both the issue of emotion regulation and psychological distress in cancer care and ePRO. The main asset of this work is the extensive and up-to-date literature review on the topics. Laypersons on, or unfamiliar people with, ER in cancer care will have an overarching picture of the models and issues of ER and methodological challenges of research into the topic. Expert people in ER in cancer care will be happy to read these new and promising perspectives. The parallel with psychotherapies assessment, and the provided references that go along with, is of particular interest. Indeed, psychosocial care in cancer settings may benefit from what is known is the field of psychotherapy. Table 2 is a good synthesis.

My small concern is the length of the paper, which may deter the entire reading. This would be really a pity as the content has to be entirely read to grasp the message well. As they are many redundancies about the necessity to think about dynamic systems in assessing and understand ER, shorten should thus be easy avoiding these unnecessary repetitions.

Here are 3 minor elements to fix:

Page 2 line 76-77. The definition of ePRO [20] is weird as it gives the impression that patients who complete data using e-devices are involved in their treatment and monitoring process. Actually, patients not involved in their treatment, but who have accepted to participate in a research (maybe for money, because bored or to make them useful), can also provide ePRO. I suggest to formulate differently the definition of ePRO.

Page 3 line 94-96. There is something wrong with the sentence “are not only based on…” we expect a “but also on….”, which is not in the sentence.

Table 1 is not clear. What is situation A and B? do you mean two different situations without any information about the situation? But I have the idea that situation A corresponds to low controllability, low interpersonal exchange, a low number of stressors but high emotional intensity and different regulatory goals. I guess that it would be better to specify “situation A : CSC with a low level (e.g. low controllability) “ and situation B  : CSC with a high level “ and to rearrange the table. Maybe it should be stated that all ER are adaptative here.

Author Response

Dear Reviewer,

Regards,

K. Mirzaie

Reviewer 2 Report

The focus of this paper is to review the literature on the role of emotion regulation in coping with distress and quality of life for children, adolescents, and young adults (CAYA) with cancer. The authors indicate the utility of using an electronic Patient Reported Outcomes (ePRO) in cancer research and care with consideration of the role of ER and proposes an alternative model to understand and study the aforementioned, as guided by the dynamic systems theory.

There are a few key areas that require clarification before this paper can be determined for suitability in the current journal. 

Major Feedback:

Broadly speaking, the authors make an effort to incorporate many conceptual frameworks, drawing from clinical psychology (psychotherapy), pediatric psychology (CAYA, ePRO), dynamic system theory, etc. Who is the intended audience for this paper? With so many theoretical and conceptual frameworks introduced, it is unclear whether this proposal may benefit researchers, clinicians, or otherwise? Further integration of the various theories/conceptual frameworks outlined is necessary to clarify the scope of this paper and what the authors hope the readers gain out of learning this new perspective.  

Structurally, the objectives of the paper can be strengthened. What type of review is offered (scoping? systematic?) and why is the alternative model proposed? What is the rationale for proposing this alternative model?  

Specific to the notion of psychotherapy and psychopathology, there are several important considerations missing from the current manuscript that needs to be addressed. The authors suggest that emotion regulation is a transdiagnostic factor/mechanism that plays a role in psychopathology. There is discussion on how emotion regulation holds implications for psychotherapy. However, this is not specified. What are the theoretical models that are considered as part of this literature (e.g., cognitive-behavioural, emotion-focused)? For example, line 180 – the authors highlight the evidence-based practice of emotional disclosure or acceptance. Are these strategies in the context of the third-wave therapies (e.g., ACT?) or another modality? Without specifying the theoretical models and how the strategies are used to target presenting concerns, it is unclear how the outlined strategies are considered evidence based.

This paper makes the argument that emotion regulation, ePRO, and an alternative model of ER (i.e., dynamic systems theory) can hold important implications for the care and research of CAYA. Although the authors highlight this point throughout the paper, it is unclear how this is the case. For example, while emotion regulation can indeed play a role in the coping of illness for young people living with and beyond their cancer, exactly how does addressing emotion regulation in this context (as the authors proposed) can inform both the care and research for CAYA? Further, the authors often make reference to broad statements – “cancer is a dynamic medical experience” – how is this the case? This was referenced very vaguely but is not clarified for how cancer is a dynamic experience and why a model such as a dynamic model of emotion regulation can be beneficial to the study and care for this vulnerable population. Offering specific examples can be helpful.

I suggest the same for discussions around ePRO – how does the use of ePRO benefit the population discussed? The authors highlight the importance of the health care provider perspective. How about those with the lived experience and how would this be useful for them?  

Building on this, the paper covers the broad spectrum of one’s cancer experience – from active treatment, to survivorship, to palliative care. I caution the authors to not overlap these experiences and/or assume their commonalities/differences. While I imagine there may be common experiences in play, there are also very distinct experiences at each phase of one’s cancer journey that requires a dedicated discussion on. Further, the authors focused on CAYA in this paper. Why is CAYA the focus and what is unique about their experience from those of adults? Developmental considerations specific to CAYA are worth noting in the manuscript. Overall,  as it stands, it is unclear how the current literature and proposed framework for emotion regulation is uniquely importantly for CAYA living with and beyond their cancer, compared to those of the general population, for instance. Therefore, it remains unclear whether Cancers is a suitable journal for this published work.

Minor Feedback:

There are many acronyms (ER, QoL, CAYA, ePRO, EFR, DST, DHS, CSM, ARC, CSC, etc). I suggest paring down the number of acronyms introduced to minimize confusion for the readers.

I suggest removing the descriptor “so-called” as this can minimize your proposed constructs/frameworks, which are important and the focus of this paper.

Section 8: Tying It All Together – Benefits and Drawbacks is quite a small section and it is unclear what the exact benefits and drawbacks are. This section may benefit from a larger summary since there is quite a bit of content to synthesize. Based on the authors’ review, what are in fact the benefits and drawbacks – high level themes should be pointed out here and details can be viewed under Table 2.

There are some conceptual distinctions that needs to be made. For example, I recommend the authors clarify the difference between psycho-oncology and psychosocial care, and then oncology and psycho-oncology.

Author Response

Dear Reviewer 2,

Regards,

K. Mirzaie

Round 2

Reviewer 2 Report

Response 3: Your clarification that this is a perspective paper integrating a critical review with personal/clinical experience makes sense and I recommend making this explicit at the onset. 

Response 4: While it is recognized that within the scope of this paper the authors are not required to discuss at length the various theoretical models that inform the strategies discussed in this paper by the authors (emotional exposure and acceptance) there is still a need to clarify the theoretical framework in which these strategies are drawn from because each strategy can be used differently depending on the model. Clinical psychology is from my understanding not a distinct field of study from psychotherapy, as psychotherapy is more broadly a term to refer to the type of psychology-based therapy provided. 

Response 5: What are the specific examples of a developmental task, and why would these tasks be unique to CAYA such that emotion regulation would be applicable to this population more than the general population? 

Author Response

Please find below the details of the revision to the manuscript with our responses to the reviewers’ and editors’ concerns and comments marked in red. The corresponding changes can be tracked accordingly in the latest revised file. All authors have read and approved the latest revised version of the manuscript.
